# Spontaneous Left External Iliac Vein Rupture

**DOI:** 10.3390/diagnostics12112820

**Published:** 2022-11-16

**Authors:** Cosmina Fugărețu, Cătalin Mișarca, Gina Vlada, Andrada Cîrnațiu, Cosmin Buzea, Daniela Marinescu

**Affiliations:** 11st General Surgery Department, Brașov County Emergency Clinical Hospital, 500326 Brașov, Romania; 2Faculty of General Medicine Brașov, Transilvania University, 500036 Brașov, Romania; 3Vascular Surgery Department, Brașov County Emergency Clinical Hospital, 500326 Brașov, Romania; 41st General Surgery Department, Emergency Hospital of Craiova, 200642 Craiova, Romania; 5Department of General Surgery, Faculty of Medicine, University of Medicine and Pharmacy of Craiova, 200349 Craiova, Romania

**Keywords:** spontaneous iliac vein rupture, spontaneous retroperitoneal hematoma, May Thurner syndrome

## Abstract

Spontaneous rupture of the Iliac Vein is very rare in practice. In over 90% of cases, the venous lesion is located on the left side. The exact etiology of this condition is unknown. Spontaneous injury of the iliac vein is thought to be favored by intense exercise, constipation, cough, labor, May-Thurner syndrome or pre-existing inflammatory changes in the venous wall are also implicated. We present the case of an 83-year-old woman who is brought to the Emergency Department for abdominal pain located in the left flank and in the left iliac fossa, which appeared after a medium physical exertion. After an emergency contrast-enhanced abdominal CT scan, the diagnosis of spontaneous rupture of the left external iliac vein is established. Surgery is performed with extreme urgency by retroperitoneal approach and due to the very precarious condition of the patient, venous ligation is done, wishing to perform a Palma-Dale venous bypass at a later time. Although a rare cause of spontaneous retroperitoneal hematoma, a non-traumatic rupture of the common or external iliac vein should be considered in patients in shock with massive retroperitoneal bleeding, accompanied by a high mortality rate.

**Figure 1 diagnostics-12-02820-f001:**
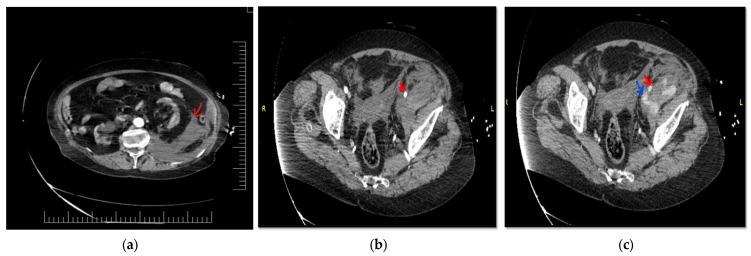
Contrast-enhanced CT scan of an 83-year-old woman that presented with intense pain in her left flank, left iliac fossa and left lower limb with a sudden onset of about 2–3 h after a medium physical exertion. When taken over by the emergency rescue service, the patient is conscious, agitated, and pale, with sweaty and cold skin. Blood pressure values are 90/50 mmHg, and the ventricular rate is 100 rpm. Personal pathological antecedents are hypertension and chronic ischemic heart disease for which it is treated with angiotensin receptor blocker, beta-blocker, thiazide diuretic and antiplatelet therapy-Aspenter-acetylsalicylic acid. Clinical examination reveals pale, cold sweaty skin, a mobile abdomen with respiratory movements, and pain in the left flank and left iliac fossa that appear swollen compared to the contralateral area, giving an asymmetrical appearance to the abdomen, without signs of peritoneal irritation. Laboratory tests show: HB-8.7 g/dL, erythrocytes 3 × 10^6^/uL, Leukocytes 10.2 × 10^3^, INR 1.55, APTT 23.3 sec, Urea 53.8 mg/dL, Creatinine 1.76 mg/dL. Blood type and Rh are determined. Intravenous fluid resuscitation is started, and the suspicion of a massive retroperitoneal hematoma is raised; a contrast-enhanced emergency CT examination is performed that confirms the diagnosis and detects a voluminous retroperitoneal hematoma of 24/12/12 cm anteriorly displacing the kidney and perirenal adipose tissue. The retroperitoneal hematoma is marked with a red arrow (**a**)**.** Contrast-enhanced CT scan during the arterial phase reveals homogeneous opacified common and external iliac arteries. It is marked with a red arrow on the left external iliac artery (**b**). Contrast-enhanced CT scan during the venous phase detects an enlarged left external iliac vein, with extravasation of contrast agent. It is marked with an interrupted blue arrow on the left external iliac vein, with contrast extravasation at this level. The continuous red arrow indicates the left external iliac artery, which is intact (**c**).

**Figure 2 diagnostics-12-02820-f002:**
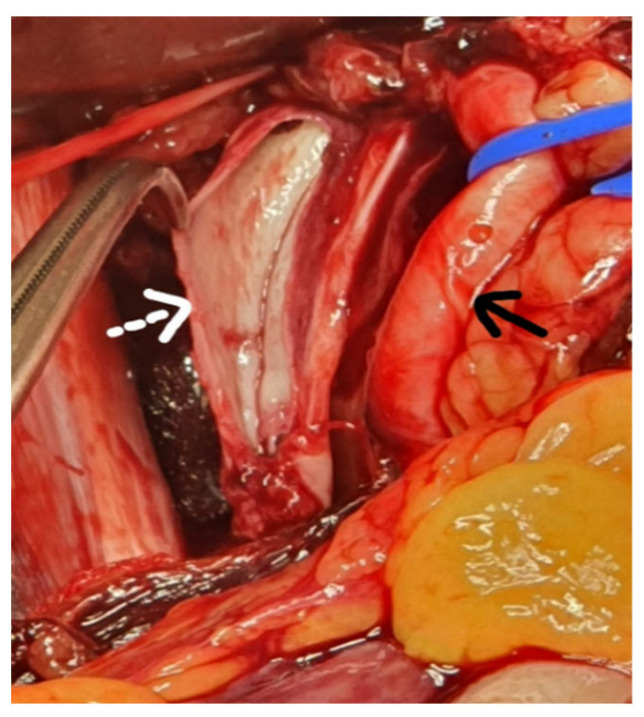
Intraoperative image. Because the patient enters cardio-respiratory arrest, the CT examination is interrupted, resuscitation maneuvers are started, and orotracheal intubation is practiced with a resumption of cardiac activity. The patient is transported directly to the operating room where emergency surgery is performed and a Leriche-Fontain incision is made with penetration into the retroperitoneum on the left side, without opening the peritoneal cavity. We opted for retroperitoneal access that facilitates the rapid identification of the iliac vessels. A large amount of aspirated blood is evacuated using a cell saver for self-transfusion. It is marked with a dashed white arrow on the left external iliac vein with a linear lesion about 4 cm long and thin venous walls. The continuous black arrow indicates the left external iliac artery, which is intact. Due to the patient’s extremely serious condition, ligation of the left external iliac vein is decided and performed above and below the lesion, with the aim of performing a Palma-Dale venous bypass at a later time. Unfortunately, the patient goes into unsustainable cardiac arrest, and intraoperative death is declared. Spontaneous rupture of the iliac vein is a very rare pathology in practice. The literature cited 53 cases until 2021, of which 14 resulted in the death of patients, with a mortality of 26.4% [1]. It seems that this pathology is much more common in women aged 41–83 years with an average of 63 years [1]. Jiang published an article in 2010 that included nine cases of spontaneous rupture of the iliac vein, with eight patients being female [2]. The most frequently affected are the common iliac vein or the external iliac vein on the left side, which accounts for almost 90% of cases, and much less often the right iliac vein. To date, only three cases of right iliac vein damage have been reported in the literature [3]. It seems that this can be explained by one of the etiological factors, namely May-Thurner or Cockett syndrome. As early as 1851, Rudolph Virchow reported in the case of corpses with left ileo-femoral thrombosis, the compression of the left common iliac vein by the right common iliac artery. In 1957, after the dissection of 430 corpses, May and Thurner showed the presence of an intraluminal fibrous band at the level of the left common iliac vein as a result of the compression made by the right common iliac artery, that was present at 22% of the corpses dissected. Cockett and Thomas are the first to report these changes to living patients [4,5]. It has been observed that spontaneous venous lesions are usually 8–40 mm long and are usually located on the anterior surface of the vein [2]. Although several theories have been advanced that try to explain the causes of this pathology, the etiology of spontaneous rupture of the iliac vein remains unknown. It is thought that changes in the venous wall, secondary to thrombophlebitis, accompanied by increased intraluminal pressure in a compressed venous segment between the inguinal ligament and the right common iliac artery, as a result of exercise or Valsalva maneuver, could lead to rupture of the venous wall [6,7]. Predisposing factors include bending and lifting exercises, defecation, labor, violent cough, and other maneuvers that can lead to increased abdominal pressure. Other authors claim that this pathology occurs more frequently in postmenopausal women because the sharp decrease in estrogen levels deprives the vascular wall of the protective effect of the latter, resulting in vascular fragility [2]. As for the differential diagnosis, it was a difficult one. Initially, rupture of an aortic aneurysm and left iliac artery was suspected, but examination of the lower limbs revealed the presence of arterial pulse to distality. Spontaneous renal rupture or Wunderlich syndrome, another cause of non-traumatic retroperitoneal hematoma, has been suspected [8]. Spontaneous rupture of the adrenal gland was taken into account, although it was found that this pathology is more common in young women [9,10]. There are similarities between the case presented and, the cases cited in the literature. The patient is female, and the symptoms appeared after a physical effort to climb some steps. In terms of age, it was at the upper end of the range found in the literature [1]. Although it is assumed that in 1/3 of cases of spontaneous rupture of the iliac vein, the May-Thurner syndrome is present, which would explain the location on the left side of the lesion, this pathology could not be documented in the present situation [11]. In front of a shocked elderly patient with pale skin, there were no signs of digestive or genital bleeding, after excluding a trauma and clinical examination of the abdomen that revealed abdominal asymmetry with significant swelling of the left flank we turned to the diagnosis of spontaneous retroperitoneal hematoma. A peculiarity of the case is the length of the venous lesion of about 4 cm, no fractures of the venous wall, no macroscopic signs of venous distress, and no signs of deep vein thrombosis. There is no consensus on the benefit of surgery in patients with retroperitoneal hematoma who are hemodynamically stable, opting in these situations for conservative treatment and dynamic follow-up of the case [12]. Although the patient presented initially responds to resuscitation maneuvers, she becomes rapidly hemodynamically unstable and emergency surgery is required. Most authors describe a trans-abdominal surgical approach, especially in cases where the source of the hemorrhage cannot be identified or there is a solution of continuity with the peritoneal cavity, the retroperitoneal hematoma being accompanied by hemoperitoneum [13,14]. We opted for the retroperitoneal approach because we considered it to be the fastest, no longer having to mobilize the left colon and possibly the small intestine to identify the iliac vessels. The incision was made at the point of maximum fluctuation, the opening of the retroperitoneum being extremely easy due to the massive hematoma, so the vessels were very quickly exposed. Some authors present it as a surgical option the venous suture per-primam, being later combated deep vein thrombosis by anticoagulant treatment and prevention of pulmonary embolism by mounting a filter on the inferior vena cava [15,16]. Considering the serious condition of the patient, we opted for the rapid accomplishment of the hemostasis, being performed the venous ligation above and below the lesion, wishing to perform at a later time a Palma-Dale type venous bypass [5]. The Palma-Dale bypass is a surgical procedure used when is an obstruction of one of the iliac veins and involves the use of the internal saphenous vein on the opposite side which remains connected to the femoral vein into which it flows and is anastomosed to the contralateral femoral vein downstream of the obstruction. In the case of a thin internal saphenous vein, an arterio-venous fistula is performed between the superficial femoral artery and the internal saphenous vein either using a saphenous collateral when possible or a 4–5 mm polytetrafluoroethylene (PTFE) prosthesis [17]. This arterio-venous fistula should be removed 3 months after surgery for optimal venous bypass function. The use of a stent placed at the level of the venous lesion was a treatment solution for a patient in Taiwan, diagnosed with spontaneous rupture of the external iliac vein [18]. Finally, we must mention that due to the lack of the traumatic factor, this case was initially managed as a major emergency by the ED doctor and not as a trauma case. He later requested the help of the general surgery on-call team for case management. The vascular surgeon joined the operating team shortly after the start of the surgery. In conclusion, a non-traumatic rupture of the common or external iliac vein should be considered in patients in shock with massive retroperitoneal bleeding, accompanied by a high mortality rate.

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
