# Peer review of "Spontaneous Left External Iliac Vein Rupture"

_diagnostics, 2022, doi:10.3390/diagnostics12112820_

Round 1

Reviewer 1 Report

Dear Authors,

You have submitted an interesting case of a patient suffering mortality secondary to cardiac arrest, stemming from hypovolaemic shock due to spontaneous iliac vein rupture. The images provided are of strong educational quality and use and these will make an excellent edition to the journal. I would be grateful if the article could be reviewed from a syntax/grammatical point of view as there are a few discrepancies here that could be addressed before publication. 

Many thanks for your submission.

Reviewer 2 Report

Very interesting case! Although a single case study the rarity of the syndrome makes it suitable for presenting. 

First, should review the technoredactation and page arrangement.  Second, I would like to see some preop pictures of the patient for the asimetrical area described and maybe a Gray-Turner sign. ( of course if the pictures are available).  Also, how did you raise the suspicion of a retroperitoneal hematoma? If the patient was in shock why perfom a ct-scan and skip over an US or direct exploration if the suspicion was already raised? ( of course judgement of the case after it happened is easier but I am curious regarding the medical judgement due to rarity of the case). What was the total time for the ambulace to arrive to the patient, home to hospital time, time of ED evaluation and time to OR? Also time to bleeding site is important and a comparisson between a median laparotomy compared to other incisions are important, also thinking to pelvis trauma bleeding where a median laparotomy is faster than any others. Did the patient had any fall before? Maybe treating the case as trauma ( with reducing times to OR) would have better outcome?  Who was leading the evaluation team? The trauma surgeon, vascular surgeon or ED doctor? Having a very organized ED with trauma bay can give a chance to this kind of patients where time to OR and time to bleeding site is more important than the surgical technique itself.  A disscussion regarding damage control surgery and other vascular techniques would be nice.   Very nice case!
